# Optimization of the Performance of Newborn Screening for X-Linked Adrenoleukodystrophy by Flow Injection Analysis Tandem Mass Spectrometry

**DOI:** 10.3390/ijns11030071

**Published:** 2025-08-29

**Authors:** Chengfang Tang, Minyi Tan, Yanna Cai, Sichi Liu, Ting Xie, Xiang Jiang, Li Tao, Yonglan Huang, Fang Tang

**Affiliations:** 1Department of Guangzhou Newborn Screening Center, Guangzhou Women and Children’s Medical Center, Guangzhou Medical University, Guangzhou 510180, China; fangfang_violet@sina.cn (C.T.); mytan2013213918@163.com (M.T.); liusichi0@126.com (S.L.); xieting01110824@163.com (T.X.); leonjpx@gmail.com (X.J.); taolliaohy_cn@126.com (L.T.); xxhuang321@163.com (Y.H.); 2Department of Genetics and Endocrinology, Guangzhou Women and Children’s Medical Center, Guangzhou Medical University, Guangzhou 510623, China; anniecai95@126.com

**Keywords:** x-linked adrenoleukodystrophy, flow injection–tandem mass spectrometry, liquid chromatography–tandem mass spectrometry, newborn screening

## Abstract

The aim of this study was to improve screening efficiency by establishing reasonable interpretation criteria for the use of flow injection analysis tandem mass spectrometry (FIA-MS/MS) in newborn screening (NBS) for X-linked adrenoleukodystrophy (X-ALD). FIA-MS/MS was employed to analyze very-long-chain acylcarnitines (ACs) and lysophosphatidylcholines (LPCs) and their ratios in dried blood spot (DBS) obtained from five X-ALD patients in the neonatal period (0–7 days old) and 7123 healthy neonate controls. By comparing these results and analyzing receiver operating characteristic (ROC) curves, we identified sensitive indicators for X-ALD screening in newborns. To evaluate the performance of different FIA-MS/MS screening indicators, we simultaneously analyzed 7712 neonatal DBS samples obtained for X-ALD screening using FIA-MS/MS and the established liquid chromatography tandem mass spectrometry (LC-MS/MS) method for quantitative detection of C26:0-lysophosphatidylcholine (C26:0-LPC). Furthermore, 84,268 newborn X-ALD screening results were retrospectively analyzed to further evaluate the screening performance of FIA-MS/MS. After the three-step optimization evaluation, the optimized first-tier sensitive screening indicators of FIA-MS/MS were C24:0-AC, C26:0LPC, and C24:0/C22:0-AC. Among the 7712 newborns screened, one case was confirmed to be double-positive. Within separate statistical analyses, based on LC-MS/MS screening alone (positive cutoff > 0.17 µmol/L), only seven cases (0.09%) were initially positive, with a positive predictive value (PPV) of 42.8%, and two additional *ABCD1* VUS hemizygous males were detected. Through the retrospective analysis of 84,268 newborns, eight *ABCD1* variants (six hemizygous males and two heterozygous females) were ultimately identified. Our study showed that the optimization of first-tier screening performance is particularly important if second-tier screening is not performed. Using LC-MS/MS for second-tier screening for X-ALD can significantly reduce the number of false positives, but the method still misses some false negatives. If it is used as a first-tier assessment, more VUS variant neonates can be detected.

## 1. Introduction

X-linked adrenoleukodystrophy (X-ALD; MIM: 300100) is the most common inherited peroxisomal disorder influencing the adrenal cortex and the central nervous system (brain inflammation and spinal cord/peripheral neuropathy). It has an estimated incidence of 1/10,500 [1]. The most common form influencing males is childhood cerebral X-ALD (CCALD), the most rapidly progressive and severe phenotype, which has an age of onset at 2 to 10 years. Hematopoietic stem cell transplantation has been confirmed as an effective treatment to halt cerebral demyelination and prevent death due to CALD in the early stages of the disease [2]. Early screening and diagnosis are crucial for diagnosing XALD in a pre-symptomatic stage [3].

In 2006, a preliminary report showed that C26:0 lysophosphatidyl choline (C26:0LPC), detected by liquid chromatography–tandem mass spectrometry (LC-MS/MS) assay, was elevated in postnatal venous dried blood spot (DBS) obtained from patients with X-ALD and peroxisome disorders compared to normal controls [4]. The assay was validated and proved to be highly sensitive and accurate in identifying individuals with a defect in peroxisomal β-oxidation, such as X-ALD [5]. A high-throughput method for measuring C20–C26 lysophosphatidylcholines (LPCs) using flow injection–tandem mass spectrometry (FIA-MS/MS) was later developed as a first-tier test to screen for X-ALD that makes use of existing equipment and expertise in Newborn Screening (NBS) laboratories [6]. In 2015, the Biochemical Mass Spectrometry Laboratory (BMSL) and Newborn Screening Quality Assurance Program (NSQAP) launched a QA program for X-ALD, which was made available to all laboratories to screen for X-ALD [7]. As part of the development of the NBS test for X-ALD and following the passage of Aidan’s Law, New York became the first state to introduce NBS for X-ALD at the end of 2013 [8]. Since then, 38 American states, Taiwan, and The Netherlands have included X-ALD in their NBS programs. C26:0-LPC was measured as an interpretation index in all NBS programs. The use of the FIA-MS/MS method to measure C26:0-LPC levels was seen as desirable, but yielded high levels of false positives. Therefore, LC-MS/MS was applied as a second-tier test in these areas, significantly reducing the false positive rate in diagnosis [9]. In Georgia, the concentrations of C20, C22, C24, and C26 LPCs, as well as the ratios between them, were used in the first tier, but with post-analytical tools (Collaborative Laboratory Integrate Reports; CLIR) being used to analyze the FIA-MS/MS results, while only C26:0LPC was used in the second tier [10]. A pilot study on the implementation of X-ALD was launched in the Italian NBS program in 2021. C20:0, C22:0, C24:0, C26:0-LPC, C20:0-, C22:0-, C24:0-, and C26:0-AC could be measured using FIA-MS/MS for first-tier testing, but with a three-tiered approach.

In Guangzhou, China, an LC-MS/MS method for the detection of C26:0-LPC was developed in our preliminary research and applied in a pilot study of NBS for X-ALD [11]. With the launch of the NeoBase 2^™^ Non-derivatized MS/MS kit in mainland China, several screening centers, including ours, have adopted the FIA-MS/MS method to include X-ALD screening as a routine screening item. However, only first-tier screening can be used at present in most screening centers around the world, and there is no framework for conducting second-tier testing. It is therefore essential to optimize the performance of first-tier test screening methods. Guo-Li Tian et al. evaluated a panel of very-long-chain acylcarnitines (ACs) and lysophosphatidylcholines (LPCs) for screening for X-ALD using FIA-MS/MS in China, but only with confirmed samples from non-neonatal children [12]. Archana Natarajan et al. hypothesized that C26:0 and C24:0-LPCs in DBS were suitable for the first-tier screening of newborns for X-ALD using FIA-MS/MS; however, C24:0 and C26:0-ACs may not be reliable markers for X-ALD screening. In addition, this method was verified using diagnosed samples from non-neonatal patients [13,14]. The current study assessed the performance of FIA-MS/MS in screening for X-ALD in newborns and established screening interpretation indicators to improve the efficiency of screening for this disease. The performances of two methods (FIA-MS/MS and LC-MS/MS) were then compared.

## 2. Materials and Methods

### 2.1. Materials

For FIA-MS/MS screening, all chemical reagents and the isotopically labeled internal standard were taken from the NeoBase 2^™^ Non-derivatized MS/MS kit (Revvity, Taicang, China). Whatman S&S 903 (Cytiva, Marlborough, MA, USA) filter paper was used for spotting blood samples and purchased from Merk, Sigma-Aldrich Corp. (St. Louis, MO, USA). For LC-MS/MS screening, the C26:0-LPC was quantified using a previously published method [11].

### 2.2. Subjects

For boxplot and receiver operating characteristic (ROC) curve analysis, five residual DBS samples were collected from confirmed X-ALD patients via routine neonatal screenings 0–7 days after birth in Guangzhou Newborn Screening Centre. DBS samples from 7123 neonates (aged between 0 and 7 days) were used as healthy controls to establish reference intervals (95% confidence intervals) of the investigated markers. The reference intervals of very-long-chain ACs, LPCs, and their ratios, as detected by FIA/MS/MS, could be defined as the value range between the 0.5th and 99.5th percentile for the control group. The positive cutoff values for these indicators were tentatively set at the upper limit of the 99.5% percentile in the population.

DBS samples from 7712 newborns born in Guangzhou from July 2023 to March 2024 were analyzed and simultaneously screened using the FIA-MS/MS and LC-MS/MS methods.

For the expanded large-population study screening for X-ALD using FIA-MS/MS (NeoBase 2™) as the first-tier screening test and screen-positive cases followed by LC-MS/MS as the second-tier test, first-tier screening test was conducted on the DBS samples of 84,268 newborns (45,345 males and 38,923 females) born in Guangzhou from July 2023 to November 2024, and second-tier test was carried out on 179 screen-positive cases.

All blood samples were collected on Whatman 903 filter paper and air-dried at room temperature for at least 4 h. The DBS samples were sealed in a plastic bag and stored at −20 °C until further analysis. This study was approved by the ethics committee of the Guangzhou Women and Children’s Medical Centre, and the authors obtained written approval for human genetic resource collection in China before sample collection from each participant.

### 2.3. FIA-MS/MS Analysis

Very-long-chain ACs (C20:0-, C22:0-, C24:0-, and C26:0-AC), LPCs (C20:0-, C22:0-, C24:0-, and C26:0-LPC), and their ratios (C24:0AC/C22:0AC, C26:0AC/C22:0AC, and C26:0LPC/C22:0LPC) were analyzed using the NeoBase 2^™^ Non-derivatized MS/MS kit (Revvity), following the manufacturer’s instructions. Single 3.2 mm disks were punched with DBSs and transferred into 96-well plates. A total of 125 μL of the NeoBase 2^™^ extraction working solution (EWS) was added to each well. The microplate was covered with an adhesive microplate cover and shaken for 30 min at 650 rpm at 45 °C. The microplate cover was removed, and 100 µL of the specimen could then be transferred to a new microplate and covered with an adhesive microplate cover before being analyzed using negative-ion-mode liquid chromatography–tandem mass spectrometry on a QSight^®^ 210MD mass spectrometer(Revvity, Taicang, China). According to our reference population, the samples would be considered potentially positive if the interpretive indicators were greater than or equal to the cutoff values, at which point the LC-MS/MS results would be combined to render a comprehensive judgment.

### 2.4. LC-MS/MS Analysis

LC-MS/MS screening for X-ALD was undertaken by measuring C26:0-LPC in DBSs using a Waters ACQUITY ultra-performance liquid chromatography (UPLC) I class system coupled with a Xevo TQD triple-quadrupole mass spectrometer (Waters, Milford, MA, USA) [11]. The samples with a C26:0-LPC value ≥ 0.17 µmol/L were categorized as positive, and the patient was referred for further testing and diagnostic evaluation.

### 2.5. Diagnosis

A positive identification by either base-FIA-MS/MS or base-LC-MS/MS screening could result in referral for confirmatory tests and follow-up. A diagnosis of X-ALD was generally considered confirmed if the very-long-chain fatty acids (VLCFAs) were consistent with this diagnosis and a known pathogenic variant (P), likely pathogenic variant (LP), or variant of uncertain significance (VUS) could be detected in the *ABCD1* gene. DBS samples from these suspicious cases and peripheral venous blood (2 mL) of related family members were obtained for subsequent genetic testing using next-generation sequencing [11].

### 2.6. Statistical Analysis

Statistical analysis was performed using SPSS 20.0 software (SPSS, Inc., Chicago, IL, USA) and MedCalc (Version 22.009). The distribution of ACs and LPC concentrations and their ratios were compared between the control and patient samples using the Mann–Whitney U test, with the significance value set at *p* < 0.05. The detection data were all skewed distributions. For non-normal distribution measurement data, quartiles [M (Q1, Q3)] or medians (range) [M (min~max)] were used. The sensitivity and specificity of biomarkers were evaluated by using ROC curves.

## 3. Results

### 3.1. Establishment of Normal Reference Range for Very-Long-Chain ACs, LPCs, and Their Ratios and Analysis of Sensitive Indicators

The concentrations of very-long-chain ACs, LPCs, and their ratios were determined using FIA-MS/MS in DBS samples obtained from five X-ALD patients and 7123 normal newborns (healthy controls). The reference ranges (95% confidence intervals and the 0.5th and 99.5th percentiles) of very-long-chain ACs, LPCs, and their ratios and the median concentration range in the patient group are presented in detail in Table 1, along with a comparative analysis using box-and-whisker plots (Figure 1 and Figure A1). As shown in Figure 1 and Figure A1, all patients had elevated C24:0AC, C26:0AC, C26:0-LPC, C24:0AC/C22:0AC, C26:0AC/C22:0AC, and C26:0LPC/C22:0LPC levels, with clear separation from the neonatal controls. ROC curve analysis was performed, and sensitive indicators were determined for ACs, LPCs, and their ratios. These were used to assess the sensitivity and specificity of our method. Table 1 presents the ACs, LPCs, and their ratios, along with their sensitivity, specificity, and area under the ROC curve (AUC) with 95% confidence intervals (CIs). These results show that the AUC values for C24:0AC, C26:0AC, C26:0-LPC, C24:0AC/C22:0AC, C26:0AC/C22:0AC, and C26:0LPC/C22:0LPC were all greater than 0.95 (*p* < 0.001), indicating a good screening performance for X-ALD.

### 3.2. Comparison of NBS for X-ALD by FIA-MS/MS Versus LC-MS/MS

A total of 7712 newborns were simultaneously screened for X-ALD by FIA-MS/MS and LC-MS/MS. Of the 7712 newborns, only one was screened as positive by both methods. A C26:0LPC level of 0.67 µmol/L (≥0.62 µmol/L), C24:0AC of 0.08 µmol/L (≥0.03 µmol/L), and an elevated C24:0AC/C22:0AC ratio of 7.8 (≥4.5) were detected by FIA-MS/MS, while a C26:0LPC level of 0.59 µmol/L (>0.17 µmol/L) was detected by LC-MS/MS. This diagnosis was confirmed with a c.1771C>T (p.R591W) hemizygous VUS variant in the *ABCD1* gene, and VLCFAs were abnormal on follow-up. The indicators of C24:0AC, C26:0AC, C26:0LPC, C24:0AC/C22:0AC, C26:0AC/C22:0AC, and C26:0LPC/C22:0LPC in FIA-MS/MS and C26:0LPC in LC-MS/MS were evaluated for use in the respective methods based on the box-and-whisker plot presented above (Table 2). For MS/MS screening, if an initial positive screening is determined based solely on elevated levels of the primary indicators C24:0AC, C26:0AC, and C26:0LPC, the number of initial positive screenings is at least 459 (5.95%), with a positive predictive value (PPV) of 0.22%. When combined with ratio-based interpretation, the top three PPV interpretation indicators were C26:0LPC and C24:0AC/C22:0AC (50%), C24:0AC and C26:0LPC (14%), and C24:0AC and C24:0AC/C22:0AC (6%).

For LC-MS/MS screening (with a positive cutoff value of >0.17 µmol/L), only seven cases (0.09%) were initially screened as positive, with a PPV of 42.8%. In addition to the same confirmed X-ALD infant described above, who was diagnosed using the two methods, two other male hemizygotes with *ABCD1* gene VUS variants were detected: one (c.670G>A (p.V224M)) with an elevated concentration of C26:0-LPC of 0.19 µmol/L based on LC-MS/MS and normal levels of C24:0AC (0.02 µmol/L), C26:0AC (0.02 µmol/L), C26:0LPC (0.44 µmol/L), C24:0AC/C22:0AC (2.88), C26:0AC/C22:0AC (2.25), and C26:0LPC/C22:0LPC (1.28) based on FIA-MS/MS, and another (c.701G>A (p.R234H)) with an elevated level of C26:0-LPC of 0.20 µmol/L based on LC-MS/MS, as well as normal levels of C24:0AC (0.02 µmol/L), C26:0AC (0.01 µmol/L), C26:0LPC (0.33 µmol/L), C24:0AC/C22:0AC (2.22), C26:0AC/C22:0AC (1.44), and C26:0LPC/C22:0LPC (1.38) based on FIA-MS/MS. When using a cutoff of 0.2 µmol/L for C26:0-LPC for LC-MS/MS, the diagnosed case became the only positive case, with a PPV of 50%.

### 3.3. Outcome of NBS for X-ALD Using FIA-MS/MS and LC-MS/MS in an Expanded Screening Population

A total of 84,268 newborns (45,345 males and 38,923 females, with a median age of 3 d) were analyzed by FIA-MS/MS as a first-tier screening test, while 179 newborns with positive initial screening results were also screened by LC-MS/MS as a second-tier test (Figure 2). If interpreted solely based on elevated C24:0AC (≥0.04 µmol/L, adjusted from 0.03 µmol/L according to Table 2), C26:0AC (≥0.06 µmol/L), and C26:0LPC (≥0.62 µmol/L) levels, the positive count from the initial screening was at least 2144 cases (2.54%). When combined with ratio-based interpretation, the number of positives in the initial screening decreased to 179 cases (0.21%). After the second-tier screening, seven cases (3.91%) were confirmed to be positive, of which six (85.7%) carried *ABCD1* variants (five male hemizygotes and one female heterozygote). The concentrations of C26:0-LPC in cases #3, #4, and #5 were all within the normal range (Table 3).

To further investigate potential false negatives, the C26:0LPC cutoff was adjusted from the 99.5th percentile to the 97.5th percentile of the population. This increased the number of positives in the initial screening to at least 6425 cases (7.62%), while with ratio-based interpretation, 327 cases (0.39%) were identified, leading to the additional detection of two *ABCD1* variants (one male hemizygote and one female heterozygote). Among these eight cases, five were recalled and tested for plasma VLCFAs; the results were all abnormal (Table 3). When only identification of X-ALD based on the *ABCD1* gene was included, the incidence of X-ALD was approximately 1 in 21,067 male live births (6 in 84,268). The most sensitive indicators of FIA-MS/MS from these cases were C24:0AC and C24:0AC/C22:0AC, and C24:0AC and C26:0-LPC.

## 4. Discussion

In many areas, NBS for X-ALD is carried out using a two- [9] or three-tier algorithm [15]. DBS samples are first analyzed using FIA-MS/MS as the first-tier test, mainly based on elevated levels of C26:0-LPC. The false positive rate in the first-tier test is much higher than in the second-tier test. The positive rate from first-tier screening was 4.2%, and that from second-tier testing was 0.02% in California [16], while this rate changed from 0.7% to 0.02% in Georgia [10]. However, many screening centers can still only perform first-tier screenings based on commercial kits, and there is no available framework for second-tier testing. If the criteria for first-tier screening methods also refer to screening centers that have conducted second-tier screening tests, that is, if it was simply based on increased C26:0LPC, the screening positive rate would be too high. Therefore, optimizing first-tier screening tests is particularly important. Researchers in India made estimates using a panel of LPCs (C26:0, C24:0, C22:0, C20:0) when screening DBS for X-ALD, finding that the C26:0 and C24:0-LPCs in DBS were suitable for first-tier screening of newborns for X-ALD; however, the age of confirmed patients with clinical validation was 3–40 years, and the patients were therefore not neonates [13]. Researchers in Shanghai also evaluated promising biomarkers of VLCFAs and LPCs to screen for X-ALD, showing that C26:0-LPC, C24, and C26 were the three most useful biomarkers for screening for this disease in children [12]. Jaspers et al. [17] demonstrated that C26:0-LPC analysis achieved a superior diagnostic performance compared with VLCFA analysis (using the C26:0 and C26:0/C22:0 ratio) in all patient groups.

To determine which screening indicators were the most valuable in first-tier newborn screening for X-ALD and to optimize screening efficiency, we first compared very-long-chain ACs, LPCs, and their ratios between confirmed X-ALD patients and healthy controls using FIAMSMS testing, while at the same time establishing a normal reference interval based on the 99.5th percentile of the population. Based on box-and-whisker plots and ROC curve analysis, sensitive indicators such as C24:0AC, C26:0AC, C26:0-LPC, C24:0AC/C22:0AC, C26:0AC/C22:0AC, and C26:0LPC/C22:0LPC were established. However, the associated criterion from our ROC curve analysis was not suitable as a cutoff value due to the small number of diagnosed samples, especially for confirmed X-ALD patients during the neonatal period. For large populations and multiple indicators, we uniformly set the positive cutoff value for these indicators to the upper limit of 99.5th percentile of the population. Combined with the results of the AUC, the only sensitive indicators that remained after this step of screening were C24:0AC, C26:0LPC, and C24:0AC/C22:0AC.

As mentioned above, LC-MS/MS is mostly used for second-tier screening for X-ALD, which means that although this method is more sensitive, it can only further reduce the number of false positives based on the positive results from the first-tier testing. What if false negatives occur in the first-tier screening? It is possible that the true positive cases have normal C26:0LPC in the first-tier screening while other ACs or LPCs are elevated, or all indicators are normal? If these cases are screened using only C26:0LPC as an indicator in the first-tier screening and then undergo second-tier screening using LC-MS/MS [1,16,18], they will be missed. The second-tier screening test would not have picked this up; therefore, some researchers have applied the LC-MS/MS method directly to first-tier screening [18,19], where higher PPVs and incidence rates have been reported. The PPV for the first-tier assay reached 67% in North Carolina, with a high incidence of 1 in 8717 births [19], but this was lower than the frequency reported in Minnesota (1 in 4845 births). Incidence rates differ considerably following the implementation of NBS in the USA, ranging from approximately 1 in 51,081 of the male population in Georgia [10] to 1 in 3878 in Minnesota [18]. In our preliminary study, the incidence of X-ALD based on first-tier test screening using LC-MS/MS was also high, at approximately 1 in 3324 male live births [11]. Therefore, one of the reasons for the higher incidence is that more VUS cases of the *ABCD1* gene have been detected by using direct LC-MS/MS methods, while those cases could not be detected by first-tier screening using the FIA-MS/MS method.

To test this theory and further evaluate whether the optimized screening indicators were reasonable or if there were false negatives in the first-tier screening test by FIA-MS/MS, we conducted a first-tier screening test using FIA-MS/MS and LC-MS/MS simultaneously on 7712 samples, rather than performing second-tier testing only after the first-tier testing was positive. The results showed that only one newborn screened positive for both methods. The most sensitive indicators were C26:0LPC and C24:0AC/C22:0AC, C24:0AC and C26:0LPC, and C24:0AC and C24:0AC/C22:0AC for first-tier screening by FIA-MS/MS alone, but when LC-MS/MS was used for first-tier screening with a cutoff value of 0.17 µmol/L C26:0-LPC, the PPV was 42.8%. In addition to the same confirmed X-ALD infant, there were two more identified VUS cases involving the *ABCD1* gene. This is consistent with the conclusion of a California study [20], in which a larger proportion of infant boys who were screened as positive were found to have a VUS on the ABCD1 gene. A possible explanation for this is that the large number of positives with VUS is the conservative second-tier C26:0-LPC cutoff (≥0.15 µmol/L) that is instituted in the beginning of California’s X-ALD newborn screening. In our study, if the positive cutoff value is 0.2 µmol/L; even if the very sensitive and specific LC-MS/MS is used as the first-tier screening method, and the PPV is 50%, these two cases would still be missed as false negatives. Therefore, we cannot blindly increase the PPV and ignore the possibility of false negative cases. However, for first-tier screening using FIA-MS/MS, these two cases would be missed no matter what. Their concentration values are much lower than the cutoff values, and even if the cutoff values are appropriately adjusted, they cannot be detected. Due to the absence of general genotype–phenotype correlations, these VUS cases of *ABCD1* gene follow-up investigations have not been described further [9]. Meanwhile, incidences of registered VUS in the *ABCD1* variant data-base have increased significantly [21], and some of these were reported as patients. Therefore, missing these VUS cases poses a significant risk when screening for X-ALD as a phenotypical late-onset disorder. Referring to international recommendations for the diagnosis and management of patients with X-ALD, in vitro fibroblast studies [22] were conducted to study the pathogenicity of a VUS in *ABCD1* and were especially recommended in asymptomatic boys and men with biomarker levels above the upper reference range of controls, but below the disease range (of the diagnostic laboratory) [3].

At present, the screening process for X-ALD in most international screening centers is that a C26:0LPC increase in the FIA-MS/MS results is regarded as the first level, while LC-MS/MS is used as the second-tier screening method. Alternatively, newborn screening for X-ALD is sometimes conducted using only a one-tier method, with continuous optimization of FIA-MS/MS indicators, meaning that VUS cases are very likely to be missed, which is very dangerous for patients. Therefore, when conditions permit, LC-MS/MS is a better choice for first-tier screening and can screen out more VUS cases without the need for further follow-up. However, it may also bring distress to some families and poses a challenge to clinicians. The Laboratory Integrated Reports (CLIR) tool might be a good way to identify some VUSs as false positives, thereby reducing the number of VUS cases and avoiding unnecessary long-term follow-up [20]. However, at present, our case management for VUSs still involves recalling all cases, performing genetic testing and counseling, determining plasma VLCFAs, and conducting regular follow-ups for about six months.

Our retrospective analysis of 84,268 newborn X-ALD screening results showed that the combination of the sensitive indicators C24:0AC, C26:0AC, C26:0-LPC, C24:0AC/C22:0AC, C26:0AC/C22:0AC, and C26:0LPC/C22:0LPC resulted in 179 positive cases based on the screening test. Among the 179 cases with positive initial screening results, there were seven positive cases and six cases with variations in *ABCD1* after second-tier test screening by LC-MSMS. Considering only identification of X-ALD based on genes, the PPV of LC-MS/MS reached 85.7%. Although the positive cutoff value for the C26:0 LPC was adjusted from the 99.5th to the 97.5th percentile, there were two new cases of genetic diagnosis, while the PPV of the first-tier testing using FIA-MS/MS decreased from 3.53% to 1.63%. The most sensitive indicators of FIA-MS/MS based on these cases were still C24:0AC and C24:0AC/C22:0AC, and C24:0AC and C26:0-LPC. Three cases (#3, #4, #5) in our study further confirmed that the C26:0LPC level during first-tier screening is not high. Even if we adjusted its cutoff value to the 97.5th percentile, the C26:0LPC of patient #3 remained normal. Centers that only conduct first-tier screening with C26:0LPC and then test this again in the second-tier screening will miss these cases. Of course, in our retrospective study, we might have missed some false negative cases, because only 179 cases were screened at the second-tier level. It is definitely not realistic for us to conduct first-tier screening with LC-MS/MS. However, if conditions permit in the future, we may use more positive samples for second-tier screening in an attempt to find more false negatives.

## 5. Conclusions

The optimization of first-tier screening methods is particularly important if second-tier screening is not carried out. Using LC-MS/MS as the second-tier screening for X-ALD can significantly reduce the number of false positives but still leads to some false negatives. If it is used as a first-tier assessment, more VUS variant neonates can be detected.

## Figures and Tables

**Figure 1 IJNS-11-00071-f001:**
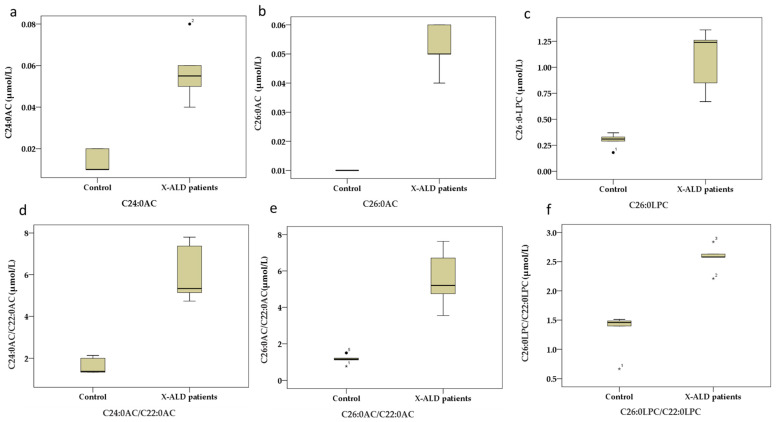
Comparison of very-long-chain acylcarnitines (ACs), lysophosphatidylcholines (LPCs), and their ratios, determined using FIA-MS/MS in X-ALD patients and healthy controls. Box-and-whisker plots depict the concentrations of very-long-chain acylcarnitines (ACs), lysophosphatidylcholines (LPCs), and their ratios (C24:0AC/C22:0AC, C26:0C/C22:0AC, and C26:0LPC/C22:0LPC) compared with healthy controls. (**a**) C24:0AC, (**b**) C26:0AC, (**c**) C26:0LPC, (**d**) C24:0AC/C22:0AC, (**e**) C26:0AC/C22:0AC, (**f**) C26:0LPC/C22:0LPC. Controls (N = 7123) and X-ALD patients (N = 5). “•” and “*”: Outlier; “number (1, 2, 3, 5)”: Patient Number.

**Figure 2 IJNS-11-00071-f002:**
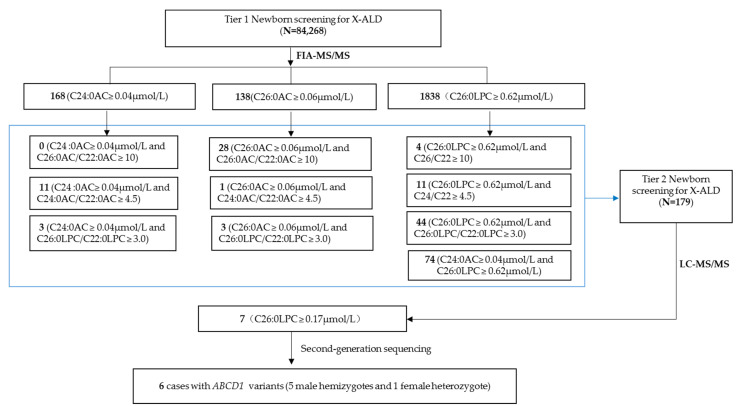
Outcomes of newborn screening for X-ALD in a retrospective analysis of 84,268 newborns born in Guangzhou (July 2023 to November 2024), using FIA-MS/MS as the first-tier screening test and, for some newborns with positive initial screening results, LC-MS/MS as the second-tier test. FIA-MS/MS: flow injection analysis tandem mass spectrometry; LC-MS/MS: liquid chromatography tandem mass spectrometry.

**Table 1 IJNS-11-00071-t001:** Concentration, sensitivity, specificity, and area under the ROC with 95% confidence interval for very-long-chain ACs, LPCs, and their ratios in controls and patients.

Controls (N = 7123)			Patients (N = 5)	Significance	Associated Criterion	Sensitivity (%)	Specificity (%)	AUC (95%CI)
Analytes	M (Q1,Q3)	Normal Reference Range [P2.5~P99.5]	M (Range)					
C20:0AC	0.02 (0.02,0.03)	0.01~0.06	0.03 (0.02,0.04)	*p* = 0.4729	>0.02	60	54.53	0.59 (0.574–0.597)
C22:0AC	0.01 (0.01,0.01)	0~0.01	0.01 (0.01,0.01)	*p* = 0.5358	>0	100	8.06	0.54 (0.527–0.550)
C24:0AC	0.01 (0.01,0.02)	0.01~0.03	0.06 (0.04,0.08)	*p* < 0.0001	>0.03	100	99.75	1 (0.999–1.000)
C26:0AC	0.01 (0.01,0.01)	0.01~0.06	0.05 (0.04,0.06)	*p* < 0.0001	>0.03	100	98.72	0.99 (0.991–0.995)
C20:0LPC	0.40 (0.31,0.50)	0.14~1.03	0.62 (0.2,0.85)	*p* = 0.0773	>0.56	80	83.56	0.73 (0.718–0.738)
C22:0LPC	0.27 (0.21,0.33)	0.09~0.60	0.44 (0.26,0.52)	*p* = 0.0073	>0.37	80	85.55	0.85 (0.838–0.854)
C24:0LPC	0.69 (0.59,0.81)	0.29~1.24	1.47 (0.62,1.80)	*p* = 0.0055	>1.06	80	97.5	0.86 (0.850–0.866)
C26:0LPC	0.32 (0.26,0.38)	0.12~0.62	1.24 (0.67,1.36)	*p* = 0.0001	>0.66	100	99.79	1 (0.999–1.000)
C24:0AC/C22:0AC	2.0 (1.64,2.57)	0.78~4.50	5.33 (4.73,7.80)	*p* = 0.0001	>4.67	100	99.66	1 (0.997–0.999)
C26:0AC/C22:0AC	1.60 (1.29,2.0)	0.61~9.39	5.20 (3.55,7.63)	*p* = 0.0002	>3.5	100	96.7	0.98 (0.980–0.986)
C26:0LPC/C22:0LPC	1.18 (0.94,1.49)	0.40~3.08	2.58 (2.21,2.84)	*p* = 0.0002	>2.2	100	96.06	0.98 (0.977–0.984)

M: median; range: [M (min~max)]; Q1: 25th percentile; Q3: 75th percentile.

**Table 2 IJNS-11-00071-t002:** Performances of the various indicators in newborn screening for X-ALD via FIA-MS/MS coupled with LC-MS/MS in 7712 newborns.

Method	Indicator (Cutoff)	Positive Screening	Diagnosed Cases	Positive Screening Rate	PPV%
FIA-MS/MS	C24:0AC (0.03 µmol/L)	315	1	4.08%	0.32%
	C24:0AC (0.03 µmol/L) and 26:0AC/C22:0AC (10)	0	0	0%	0
	C24:0AC (0.03 µmol/L) and C24:0AC/C22:0AC (4.5)	17	1	0.22%	6%
	C24:0AC (0.03 µmol/L) and C26:0LPC/C22:0LPC (3.0)	1	0	0.01%	0
	C26:0AC (0.06 µmol/L)	35	0	0.45%	0
	C26:0LPC (0.62 µmol/L)	55	1	0.71%	1.81
	C26:0LPC (0.62 µmol/L) and C26:0AC/C22:0AC (10)	2	0	0.03%	0
	C26:0LPC (0.62 µmol/L) and C24:0AC/C22:0AC (4.5)	2	1	0.03%	50%
	C26:0LPC (0.62 µmol/L)and C26:0LPC/C22:0LPC (3.0)	5	0	0.06%	0
	C24:0AC (0.03 µmol/L) and C26:0LPC (0.62 µmol/L)	7	1	0.09%	14%
LC-MS/MS	C26:0LPC (0.17 µmol/L)	7	3	0.09%	42.80%
	C26:0LPC (0.2 µmol/L)	2	1	0.03%	50.00%

FIA-MS/MS: flow injection analysis tandem mass spectrometry; LC-MS/MS: liquid chromatog-raphy tandem mass spectrometry. PPV: positive predictive value.

**Table 3 IJNS-11-00071-t003:** Neonatal two-tier screening results, plasma VLCFAs, molecular features, and family histories of eight infants with *ABCD1* variants, identified among 84,268 neonates.

No.	#1	#2	#3	#4	#5	#6	#7	#8
Sex	Male	Female	Male	Male	Male	Female	Male	Male
Age	2 d	2 d	3 d	4 d	3 d	2 d	3 d	4 d
FIA-MS/MS(NBS2)								
C24:0AC (<0.04) µmol/L	0.08	0.04	0.05	0.05	0.04	0.05	0.06	0.05
C26:0AC (<0.06) µmol/L	0.05	0.04	0.04	0.04	0.03	0.03	0.06	0.03
C26:0LPC (<0.62) µmol/L	0.67	0.95	0.39	0.54	0.56	0.58	0.61	0.46
C24:0AC/C22:0AC (<4.5)	7.8	3.91	4.91	4.5	2.77	3.25	5.17	5
C26:0AC/C22:0AC (<10)	5.2	3.18	3.45	3.42	1.92	1.69	4.58	3
C26:0LPC/C22:0LPC (<3.0)	2.58	3.21	1.39	0.78	1.72	1.76	2.42	1.12
LC-MS/MS								
C26:0LPC (<0.17) µmol/L	0.59	0.73	0.27	0.22	0.42	0.34	0.5	0.2
ABCD1 Gene								
Exon	Exon7	Exon1	Exon9	Exon7	Exon1	Exon2	Exon2	Exon1
Nucleotide change	c.1771C>T	c.406C>T	c.1915G>A	c.1736T>C	c.839G>A	c.979T>G	c.1028G>C	c.318C>G
Protein Change	p.R591W	p.Gln136	p.Val639Met	p.Ile579Thr	p.Arg280His	p.Tyr327Asp	p.Gly343Ala	p.Phe106Leu
ACMG category	VUS	LP	VUS	VUS	P	VUS	VUS	VUS
Inheritance	Maternal	Maternal	Maternal	Maternal	Maternal	-	-	-
Plasma VLCFAs								
C22:0 (<104.3) µmol/L	25.66	50.52	48.32	35.82	60.52	-	-	-
C24:0 (<94.3) µmol/L	43.86	59.78	45.04	38.16	51.16	-	-	-
C26:0 (<0.89) µmol/L	2.82	2.98	1.58	1.14	1.84	-	-	-
C26/C22 (<0.013)	0.11	1.18	0.932	1.065	0.845	-	-	-
C24/C22 (<1.04)	1.709	0.059	0.033	0.032	0.03	-	-	-
Family history	No	Uncle	No	No	No	-	-	-

“-”: denotes that the clinical information was unavailable due to loss of follow-up. d = day. P: pathogenic. LP: likely pathogenic. VUS: variant of uncertain significance. FIA-MS/MS: flow injection analysis tandem mass spectrometry; LC-MS/MS: liquid chromatog-raphy tandem mass spectrometry. VLCFAs: very-long-chain fatty acids. Inheritance pattern is provided for families in whom parental testing was performed and where the results were available. Known family history is denoted for patients in whom there were family members with a confirmed, symptomatic X-ALD phenotype.

## Data Availability

Data will be made available to qualified researchers on request.

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
