# Peer review of "Optimization of the Performance of Newborn Screening for X-Linked Adrenoleukodystrophy by Flow Injection Analysis Tandem Mass Spectrometry"

_2409-515X, 2025, doi:10.3390/ijns11030071_

Round 1

Reviewer 1 Report

Comments and Suggestions for Authors

Introduction:

Line 46:  is this incidence overall or males only?

Line 62-64:  This should be in the past tense

General introduction:  does not adequately address the need for second tier testing to reduce FP results observed by most commonly used first tier methods, and the magnitude of the issue.  This is particularly important as the manuscript is presenting a single-tiered FIA algorithm, but many attempts at this have been tried and not been successful.

Starting line 78:  Why could the second tier or first tier LC method initially developed not continue to be used for screening.

Line 98:  are all patients included in this study as confirmed cases males in both age groups?

Line 125 and line 128:  should this be uL (microliters) not mL?

Line 144:  should this be “variant of uncertain significance”

Line 153:  should this be controls, or is it intended to refer to the reference population?

Throughout the article, it is challenging to keep track of what analytes are being referred to when both acylcarnitines and LPCs are being discussed in the same sentence or short period of time in the text.  This is particularly confusing when ratios are being discussed.  Suggest using “AC” for the acylcarnitines at all times to make it explicitly obvious which species are being discussed.

Figure 1:  The legends are a bit fuzzy, even when reviewing the PDF electronically.  This makes it difficult to review the significance of the findings on the figures.  In the caption of Figure 1, it refers to VLCFA and LPCs originating from NeoBase2.  Neobase2 does not measure VLCFA, it measures the acylcarnitine and LPCs that correlate to the VLCFA.  The caption needs to be adjusted (see previous paragraph of suggestions as well). 

For the C26 acylcarnitine (pane 1B), why is the cutoff set above all the patients of newborn age?  Similarly for panes 1D, 1E and 1F.  This generally shows the need for an additional tier of testing, as setting a cutoff based on the first tier control values, show high levels of overlap between the normal and true positive populations.

For Table 1, when there are multiple analytes listed in a row, is this an algorithm that requires all listed analytes to be elevated or is there any other logic used?  The comma used is not clear.  Consider ‘+’ or AND to make it clear.

For Table 2, consider making this table horizontal, as the formatting right now is difficult to follow

In the discussion and conclusions, this needs to be simplified and clarified, as it is unclear what the end summary is.  Additionally, these conclusions are drawn from a very small number of patients, and there may be dangers with generalizing from this small data set over a tight time frame.

Reviewer 2 Report

Comments and Suggestions for Authors

The study addresses an urgent demand for X-ALD newborn screening better fit for situations where only first-tier FIA-MS/MS was available, which is of considerable relevance in view of varying resources between screening centers, particularly in China and similar situations.

I have a few concerns and suggestions to the authors:

Major issues

 1-Sample size: The comparison of biomarker levels and selection of cutoff values is initially based on five X-ALD patients (neonates), five older patients (8–10 years), and population controls.This is a very small number for deriving screening parameters, and may not fully capture the biological variability in the population. Conclusions about the most sensitive indicators in neonates would be strengthened by more cases or validation with additional cohorts.

I suggest the authors increase the sample size for newborn or state the limitation of the sample size clearly.

2- Potential for False Negatives Not Fully Addressed: While the efforts are made to improve PPV thus reducing the number of false positives, they authors also acknowledge the possibility of having “false negative” in the selected first-tier criteria yet did not elaborate on the issue.– As X-ALD is a peroxisomal disorder that can have variable elevation of the selected marker in neonatal period, by having more stringent cutoffs might miss previously reported borderline or outlier cases.

Of the 7,712 samples screened, only a single case was positive by both FAI-MS/MS and LC-MS/MS.Two additional “VUS” cases detected by LC-MS/MS were missed by the first-tier.Long term risk of missing true X-ALD cases is not well understood. The authors should discuss this issue.

3- VUS Reporting and Clinical Implications: The authors repeatedly comment on detection of ABCD1, Variants of Uncertain Significance (VUS) among the screen-positives, particularly in the scenario when LC-MS/MS is used as the first-tier.However, they do not describe the clinical work-up and follow-up plan of these VUS cases.

The discussion would be strengthened by greater discussion of the impact of VUS reporting on families, clinicians, and long-term follow up, as these are the real-world implementation challenges. Some VUS might be causing the disease. I strongly suggest the authors to xxpand the discussion on how VUS in the ABCD1 gene should be managed, including the pathways for further biochemical/functional studies, genetic counseling, and family follow-up.

4- Generalizability and applicability: The optimized markers and cutoffs are based on the specific population and instruments in the authors’ center, and the NeoBase 2™ kit.There is no indication of how those criteria may generalize to other populations or MS/MS platforms, given reported variability between laboratories in the quantifications of very long-chain fatty acids.

5- Statistical analysis: There is a lack of detailed statistical analysis (e.g., ROC curves, sensitivity/specificity of different cutoffs in the large population), frequently included in optimization of screening algorithms.

I suggest the authors to perform ROC curve analyses.This analysis displays the trade-offs between sensitivity and specificity for a test, over a range of cutoff values, and also provide 95% confidence intervals for PPV, NPV and other performance characteristics.

Minor issues:

The text would be improved by including more schematic figures or flowcharts to better illustrate the screening algorithm and decisions making process in a more intuitive manner.

Certain wording in the methods is unclear or redundant (e.g.confirmed case, screen-positive). The authors should use standardized gene, disease, and biomarker names without fail and with a view to the international readership.

In the discussion, they may offer guidelines for the reporting and follow-​​up of VUS cases in order to minimise family anxiety and allow early recognition.

Comments on the Quality of English Language

The manuscript should be revised for English grammar and clarity to avoid confusion, especially in the methodology and results sections.

Round 2

Reviewer 1 Report

Comments and Suggestions for Authors

Thank you for the thorough response to the initial review.  All of my concerns have been addressed.

Author Response

Thank you very much.  And the figures in the article have been modified.

Reviewer 2 Report

Comments and Suggestions for Authors

lined 389 screened should be changed to detected. 

Author Response

Thank you. lined 389 "screened" has been changed to "detected" ,and it has also been modified simultaneously in the abstract. The figures in the article have been modified and the manuscript also have been re-edited by Author Services Coordinator, MDPI.